# Review of Obstacle Detection Systems for Collision Avoidance of Autonomous Underwater Vehicles Tested in a Real Environment

**Rafał Kot** [ID]

**Abstract:** The high efficiency of obstacle detection system (ODS) is essential to obtain the high performance of autonomous underwater vehicles (AUVs) carrying out a mission in a complex underwater environment. Based on the previous literature analysis, that include path planning and collision avoidance algorithms, the solutions which operation was confirmed by tests in a real-world environment were selected for this paper consideration. These studies were subjected to a deeper analysis assessing the effectiveness of the obstacle detection algorithms. The analysis shows that over the years, ODSs being improved and provide greater detection accuracy that results in better AUV response time. Almost all analysed methods are based on the conventional approach to obstacle detection. In the future, even better ODSs parameters could be achieved by using artificial intelligence (AI) methods.

**Keywords:** obstacle detection; collision avoidance; path planning; image processing; autonomous underwater vehicle





## 1. Introduction

The efficiency and accuracy of the obstacle detection systems (ODSs) are strictly dependent on the parameters of the equipment and devices used. In recent years, significant technological progress has been made in this field, including increasing the accuracy and speed of the operational devices and perception sensors as well as increasing the efficiency of computing systems. The air, ground, and underwater environment presents different characteristics and parameters of signals' attenuation, reflection, and propagation, so that the hardware setup solution must be adjusted to the environment in which the ODS is expected to operate.

ODS is an essential element of the autonomous underwater vehicles (AUVs), allowing collision-free movement in an unfamiliar environment in the presence of obstacles. They are also a necessary component of path planning and collision avoidance systems or high-level controller of AUV [1]. Efficiency of ODS has a decisive impact on the operation speed and decision-making about the movement of the AUV in the event of an obstacle. The ODS workflow from the detection of an obstacle to its avoidance maneuvers, is presented in Figure 1. The initial stage includes pre-processing and environment detection. At this stage, the selection of the environmental perception sensor (e.g., sonar, camera, echosounder, laser scanner) and the appropriate tuning of the detection parameters have a key importance for the final parameters of the ODS. Next, image processing steps so-called image segmentation and morphological operations are performed. Based on the collected data from the above-mentioned procedures, the AUV path of movement is determined. Using the implemented AUV collision avoidance algorithms, the vehicle moves in accordance with the designated path, performing collision avoidance maneuvers.

Despite many simulation-tested methods of collision avoidance and path planning ensuring very good operating parameters [2], only a few solutions have been integrated

with obstacle detection systems and tested in AUV in a real-world environment. The low number of real-world-environment tests is due to its necessity to properly prepare systems implemented in underwater vehicles and its testing of interoperability. It is usually time-consuming and must be preceded by an appropriate analysis related to the influence of the environment on the tested object. Designing an AUV that is operating and making decisions in near real-time together with moving along the optimal path is still a big challenge. It requires the use of data processing algorithms and very efficient computing systems. The data received from the environment perception devices must be appropriately processed to ensure quick and accurate environment imaging, obstacle detection, their classification, and determining a safe path.

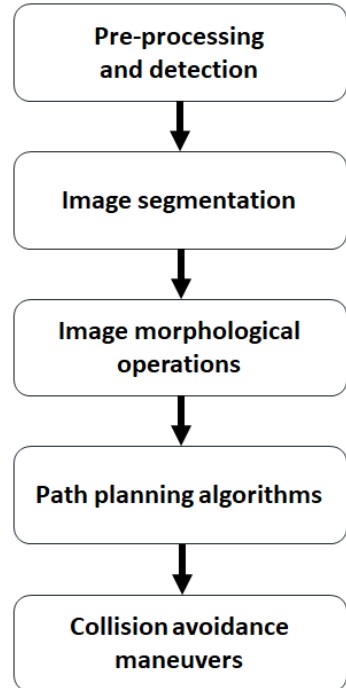

**Figure 1.** Workflow of ODS operations in conjunction with path planning and collision avoidance algorithms. The image of the environment obtained as a result of the detection is processed using image processing techniques. Then the AUV movement path is determined depending on the position of obstacles and collision avoidance maneuvers are performed, taking into account the design constraints of the vehicle.

Because of the rapid development of obstacle detection methods, attempts are systematically made to summarize and assess the state-of-the-art in this field. For autonomous vehicles (AVs) and unmanned ground vehicles (UGVs) the description of progress in this field was presented in references [3–6]. For unmanned aerial vehicles (UAVs) vision-based obstacle detection algorithms were analysed in references [7,8]. For unmanned aerial systems (UASs) obstacle detection algorithms based on deep learning (DL) were presented in [9]. Deep learning-based techniques for obstacle detection and target recognition for unmanned underwater vehicles (UUVs) have been analysed in [10–12]. Underwater mine detection and classification methods for UUVs and unmanned surface vehicles (USVs) have been discussed in [13]. In the reference [14] passive, active and collaborative detection technologies were discussed. The main surveys connected with obstacle detection methods for robots are presented in Table 1.

Table 1. The main surveys connected with obstacle detection methods for robots.

| Source | Year | Field of Analysis | Main Focus |
|---|---|---|---|
| [3] | 2014 | Ground robots (UGVs) | Image processing, obstacle detection, and collision avoidance algorithms for UGVs. |
| [7] | 2017 | Aerial robots (UAVs) | Vision-based applications for UAVs including issues such as visual odometry, obstacle detection, mapping, and localization. |
| [9] | 2019 | Aerial robots (UASs) | Deep learning methods for UASs in the context of obstacle detection and collision avoidance. |
| [11] | 2020 | Marine robots (UUVs) | Recent development of applying deep learning algorithms for sonar automatic target recognition, tracking, and detection for UUVs. |
| [12] | 2020 | Marine robots (AUVs) | Application of deep learning methods in underwater image analysis and description of the main underwater target recognition methods for AUV. |
| [4] | 2020 | Ground robots (AVs) | Advancements of obstacle detection systems for AVs. |
| [5] | 2020 | Ground robots (AVs) | AVs development regard to obstacle detection and track detection. |
| [6] | 2021 | Ground robots (AVs) | Review of the obstacle detection and avoidance approaches for AVs. |
| [13] | 2021 | Marine robots (UUVs, USVs) | Underwater mine detection and classification techniques based on sonar imagery and the classical image processing, machine learning, and deep learning methods. |
| [14] | 2021 | Marine robots (UUVs) | Data acquisition technology in underwater acoustic detection field with regard to the passive detection, active detection, and collaborative detection technology. |
| [8] | 2022 | Robots (UAVs, AVs) | Vision-based obstacle detection algorithms mainly for UAVs and also for Autonomous Vehicles. |
| [10] | 2022 | Marine robots (AUVs) | Deep learning approaches for automatic target recognition (ATR) equipped with side scan sonar and synthetic-aperture sonar imagery for AUVs |
| **This review** | **2022** | **Marine robots (AUVs)** | **Obstacle detection systems integrated with path planning and collision avoidance systems in AUVs tested in a real environment.** |

This article presents an analysis of the ODSs implemented in the AUVs with ability to avoid collision and path planning. Based on the previous literature analysis, including the path planning and collision avoidance algorithms discussed in [2], studies that presented their operation testing in a real-world environment were selected for this review. These solutions will be subject to deeper analysis in termsof assessing the effectiveness of the obstacle detection algorithms. It should be noted that only a few publications are presenting underwater ODSs integrated with path planning and collision avoidance systems. Many publications only provide path planning and collision avoidance methods tested in a simulated environment [15–17]. Some authors tested the operation of the AUV in real-world conditions where the obstacles are generated numerically [18]. Also, many authors focus only on presenting the ODS [19]. For the efficient operation of an AUV, the integration of these three systems is essential, ensuring fast and effective data processing and decision making. It also should be mentioned that among the articles related to aerial, ground, and underwater unmanned vehicles, a significantly low number of publications specifically for the underwater environment is noticeable. Most papers focus on solutions for ground or aerial environments.

The paper is organized as follows. Section 2 discusses basic technical knowledge on the sensors used to perceive the environment in which the AUV operates, and Section 3 describes the image processing steps. Section 4 provides a detailed assessment of obstacle

detection systems integrated with path planning and collision avoidance systems in AUVs tested in a real environment. Section 5 summarizes the analysis performed and identifies the most crucial problems in obstacle detection, development constraints, and future works. The final conclusions are presented in Section 6.

## 2. Perception Sensors in Underwater ODS

Specific parameters of the underwater environment, such as strong attenuation of high-frequency signals and limited access to light, have a significant impact on the choice of the environment perception sensor in operations. The most common devices for the perception of the underwater environment are sensors based on a hydroacoustic wave, such as side scan sonar (SSS), single/multibeam echosounder (SBES, MBES), or forward looking sonar (FLS). Underwater cameras are also increasingly used. In addition, UUV can be equipped with other devices, such as a laser scanner or a combination of the above sensors. There are also passive solutions for obstacle detection [20–23]. However, despite the advantages of hydrophones, such as low consumption, high concealment, and long working time, such systems only provide information about the presence of an obstacle and the signal direction of arrival (DOA) from the object. In the case of obstacles that do not carry any transmission, detecting the object is impossible. Additionally, these types of methods are very susceptible to interference. Therefore, when the AUV navigates in a complex environment, such methods have limited capabilities and can only be used as auxiliary systems. In this section, the most commonly used sensors of environmental perception will be discussed with their principles of operations, advantages, and disadvantages, limitations, and examples of applications.

### 2.1. Sonar

Sound propagation in the underwater environment can be represented by the plane wave equation (depending on spatial variations in pressure and time) as below [24]:

$$p(x,t) = A sin(2\pi f t - \frac{2\pi x}{\lambda}) = A sin(\omega t - kx) \tag{1}$$

where:
$A$—peak amplitude of the acoustic pressure of a plane wave,
$f$—frequency in Hz,
$\lambda$—wavelength,
$\omega$—angular frequency in rad/s,
$k$—wave number in rad/m.

The transmitted beam takes the shape of a cone. Its aperture angle is determined by the physical parameters of the sonar device. The basic equation representing the performance of the sonar, taking into account the parameters of the object, the environment, and the transducer, is shown below [25].

$$SE = (SL + TS - 2PL) - N - DT \tag{2}$$

where:
$SL$—source level
$TS$—target strength
$PL$—propagation loss
$N$—noise level
$DT$—detection threshold

Based on the above equation, it is possible to calculate the transmitter power needed to detect an object of a given size at a known distance. The equation shows the relationship for the FLS signal but is also true for the echosounder and SSS. A broader analysis of the acoustic signals and their underwater propagation can be found in references [26–29]. The difference between devices such as FLS, SSS, and echosounder is usually the shape of the beam, range, frequency, number of pulses, etc.

The principle of sonar operation is to send a short-sounding pulse called "ping" in the form of a hydroacoustic wave with specific parameters to explore the space. The transmitted probe signal partially reflects off the object in the range of the wave and goes back to the receiver. The detected object distance is calculated from the sound propagation time underwater according to the following equation:

$$R = \frac{ct}{2} \tag{3}$$

where:
$c$—speed of sound in an underwater environment,
$t$—time for the sound to reach the target and back to the transducer.

An echo intensity image is created based on the information about the distance and power of the signal reflected from the objects in irradiated space. The sonar signal may have different parameters and modulations depending on the expected accuracy, resolution, and speed of operation. One type of sonar signal is a fixed-frequency pulse that typically lasts several tens of microseconds [30]. The pulse duration is extended to achieve the required range without increasing the transmitter's peak power. It reduces distinguish ability in the distance [31], which means that in the case of two objects located close to each other, the echo of both objects may be combined into one reflection. Additionally, this type of signal is susceptible to interference. Avoiding this phenomenon is associated with the reduction of the pulse duration. However, if the peak power of the pulse is not increased, it will reduce the sonar range. Therefore, increasing distance discrimination and range is typically accomplished by applying signal compression. In this solution, the pulse is frequency or phase modulated. Modern sonar for AUVs uses a signal called "chirp" (Figure 2), where the frequency increases as a function of time. For example, the Tritech SeaPrince sonar's frequency varies from 500 kHz to 900 kHz [30]. The receiving of the pulse reflected from objects in space is realized using matched filters. As a result of the correlation analysis, it is possible to interpret the distance at which the object is located precisely. Additionally, an increase in the signal to noise ratio (SNR) is obtained. Such a structure of the ping makes the distance discrimination from the sonar range independent.

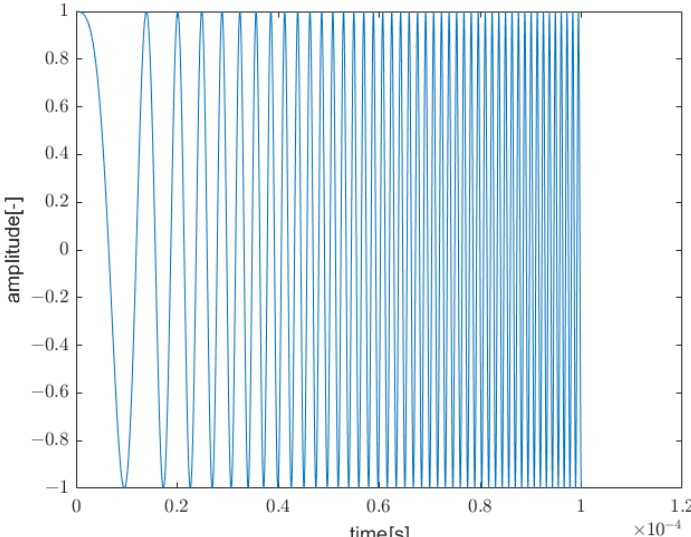

**Figure 2.** An example of a frequency-modulated signal called chirp. This type of sampling signal is commonly used in high-precision sonars. The different frequency at each point of the pulse provides high distance resolution and low susceptibility to interference.

Typical sonar solutions for small AUVs and remotely operated vehicles (ROVs) use mechanical scanning sonar (MSS). In this type of sonar, after examining the space on a

specific bearing, the transducer head changes the transmission angle mechanically, and then the next part of the space is probed [32]. The user can set the resolution, range, sector, and in some solutions, also scanning frequency, which directly affects the scanning time. In reference [33], MSS was used for the online processing framework for multiple-target tracking in cluttered environments.

Multibeam sonars are typically used as FLS in AUV. Mounted permanently at the front of the vehicle, they provide detailed information about the environment in front of the vehicle by sending multiple (several to several hundred) narrow beams in a limited field of view (FOV).

Side Scan Sonar requires the movement of the vehicle or platform on which it is mounted to change a part of the scanned space. A single ping or several pings are sent toward the object, creating a portion of its display. Repeated pulses sent from a moving vehicle generate an image of the seabed. Its primary purpose is to map the seabed or glaciers [34,35].

The single or multibeam echosounder [36,37] can also be used to measure the seabed parameters. Based on the sonar signal, SBES/MBES determines the distance to the seabed or an object within its range. The MBES generates many single narrow beams covering a larger space than an SBES and provides higher imaging resolution [38]. Due to the narrow beam, echosounders can be used as part of a simple ODS mounted on the front of the AUV operating in shallow water bodies or near the bottom/water surface. In the article [39], the echosounder, in conjunction with the depth sensor, was used to calculate the calibration factor of the vision system. The advantage of using sensors based on a hydroacoustic wave is a high distance resolution and a relatively large range that can be obtained thanks to the low attenuation of sonar signals underwater. In practice, because of reflections from the bottom and the water surface, it is necessary to limit the sonar range to several dozen meters, depending on the depth of the reservoir in which the measurements are carried out. Another disadvantage is the disturbance related to the multipath propagation of the hydroacoustic signal. Additionally, obtaining 3D images requires extensive computational resources.

*2.2. Camera*

Underwater cameras, just like ordinary cameras, capture the light reflected from the elements of the environment towards the lens. Through a matrix of photosensitive elements called pixels, an image depending on light intensity is saved in a digital form. The more pixels in the matrix, the higher the image resolution. The light colors are appropriately filtered for individual pixels to obtain three matrices corresponding to red, green, and blue color intensity. This method of color reproduction is called the RGB system and is widely used in visual image processing. Nowadays, digital cameras can record images with very high resolution and wide viewing angles. Thanks to this, the object's image is richer with many visual features that are impossible to achieve in sonar imaging [40]. Based on such features as color, the intensity of individual RGB colors, contrast, texture, and contours, it is possible to determine the size of the obstacle and its type more accurately. The disadvantage of a high-resolution video image is the need to process a large amount of data, which requires a significant computing power of ODS. Reducing the resolution causes many details to be lost at the cost of faster image processing. Another disadvantage is the need for light in the registered environment. In the underwater environment, the light intensity decreases with increasing depth. Dirty water can cause additional restrictions on incoming light. Even when the water is clean, the visibility does not exceed several meters because light rays are absorbed and converted into heat [38]. These factors also have a direct impact on the operating range of the system. For this reason, the perception of the underwater environment is highly dependent on the clarity of the water and operation depth. The camera-based ODS will also fail in under-ice missions. Despite the disadvantages, systems based on a visual image show effective performance in detecting and tracking cables located at the bottom of water reservoirs [41]. In reference [42], an image from the camera was used to determine

the trajectory of swimmers in the pool and in reference [43] to swim-fins efficiency analysis. The visual image is also chosen in object classification [44], object recognition [45] and target recognition based on deep learning methods [46]. In reference [47] the stereo vision system was used to control inspection-class ROV. Vision-based system providing position data in coastal areas was proposed in [48].

For a comprehensive solution that will guarantee high accuracy and efficiency, it is necessary to use additional sensors such as a depth sensor or a distance measurement sensor, e.g., an echosounder used to scale the visual image.

### 2.3. Other

Despite the dominance of sonar systems, other obstacle-detection solutions in the underwater environment are also being tested. An example is the laser scanner system based on a laser beam to illuminate the space in front of the vehicle. Usually, a camera is used in tandem with a laser. If there is an obstacle within the laser's operation, a strong reflection is visible in the image captured by the camera. It allows for determining where the object to be avoided is. The disadvantage of this solution is its relatively short range. The [19] presents an ODS based on a 3D image reconstruction Laser Line Scan method cooperating with the binocular stereo-vision system. In a study carried out in stable conditions in a laboratory water reservoir, the effectiveness of the object reconstruction for various conditions of water turbidity was investigated. It was found that the error in determining point clouds in water significantly increased compared to air.

Another solution can be using different combinations of sensors, such as a laser camera and sonar [49]. However, it should be noted that excessive data requires appropriate computing power to use the devices' potential. Additionally, it is necessary to consider the mutual interference of sensors, such as a combination of FLS and several echosounders, the operating bands of which may overlap. Then it is required to synchronize the operation of the devices, which reduces the speed of the AUV control system. This type of interference does not occur when combining vision and sonar systems.

### 2.4. Summary

When analysing the advantages and disadvantages of the above solutions, sonar seems to be the most appropriate sensor for the underwater environment. Depending on the intended use and the conditions in which it will operate, a device should be selected with parameters ensuring proper operating efficiency. Sensor combinations also is considered as an effective solution. A prerequisite is the appropriate computational performance of the ODS needed to process the data from these sensors.

## 3. Image Processing in ODS

This section discusses the basic image processing operations related to the classical obstacle detection process. It should be noted that the scheme of obstacle detection in AI methods is different. Each step in the classic sense of the problem corresponds to the other operation in, e.g., a convolutional neural network (CNN) based scheme. This article focuses mainly on the classical approach to image detection and processing of underwater objects.

The raw sonar data includes the echo of the reflected signal and the noise interference Figure 3. Filtering out this noise is crucial for proper obstacle detection. For this purpose, sonar data is processed using methods such as mean or median filtering, a histogram for local image values (e.g., $5 \times 5$ pixels), threshold segmentation, and filtration in removing groups of pixels smaller than $a \times b$ pixels window. The next step in sonar image processing is the image morphology process. In this case, methods such as edge determination, template matching, dilation, erosion, and combinations of the above techniques are used. After such processing of the sonar image, the obstacles' characteristics can be determined in detail, and false reflections can be avoided. Processing a high-resolution image increases the computation time needed to determine the features of the detected objects. Therefore, selecting the correct sonar resolution and image processing methods is crucial. In the

case of vision, image processing is based mainly on visual characteristics such as color, contrast, and the intensity of individual colors. The above features are also crucial in object detection techniques based on machine learning. The use of AI methods in obstacle detection systems seems very promising and prospective. The development of this type of method began about ten years ago when the authors of [50] presented an extensive deep convolutional neural network called AlexNet to recognize objects in high-resolution images with outstanding results. Since then, many modifications have been made to improve the speed and accuracy of obstacle detection and object features using CNN. The advantage of this type of solution is the high efficiency and precision of identifying and classifying obstacles, which is not comparable to classical methods. The condition is the appropriate adaptation of the neural network consisting of training with images of real obstacles. Supervised learning is very time-consuming. Additionally, it is uncertain whether, if CNN has been trained to detect, e.g., mines or submarines, it will be equally good at dealing with other obstacles that have not been trained. Another problem is acquiring a large enough quantity of training materials that DL methods guarantee a very high level of performance. Due to the limited training material and the quality of available sonar images, detecting and classifying obstacles based on deep learning algorithms are not very developed for sonar images [51]. Choosing inappropriate training resources can have a negative impact on training results.

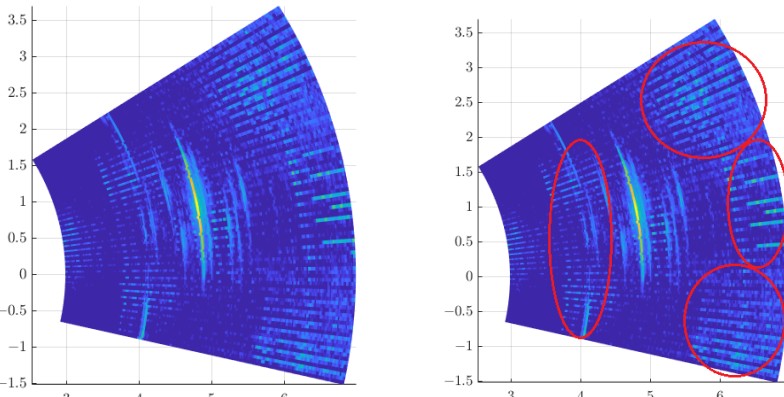

**Figure 3.** Examples of noise in sonar imaging. The red marks highlight the disturbances caused by reflections in combination with the high angular resolution of the space scan. To avoid this type of interference pre-processing operations must be performed, together with the tuning of the detection parameters. [own source].

### 3.1. Pre-Processing and Detection

At this stage, depending on the purpose of the obstacle detection method, various operations may be performed. The main objective is to avoid disturbances at the detection stage or to reduce disturbances generated during the detection. Pre-processing algorithms prepare the image for further image-processing steps. In the pre-processing and detection stage, methods based on white balance, color correction, and histogram equalization are used in camera images. For example, the study [52] presents an algorithm based on contrast mask and contrast-limited adaptive histogram equalization (CLAHE), which improves the image by visualizing the details of the object and compensates for light attenuation in captured images in an underwater environment. CLAHE was also used in [39] for video image processing in the pre-processing step in the simultaneous localization and mapping (SLAM) system. At this stage, the image can be divided into individual matrices containing grayscale with color intensity in the RGB system [53]. In the case of mine detection, the preparation of the image requires prior determination of shaded areas, areas of reflection from the bottom and water surface, and reflections from the object [54]. The image is pre-normalized to reduce noise and distinguish the background from the highlight and shadow of the mine by, e.g., using the histogram equalization operation [13].

In references [55,56], median filtering was used as part of pre-processing for sonar data, which consists in ordering adjacent pixels and then applying the median value for a specific filter size. The authors of the references above chose a window size of $5 \times 5$ pixels. The mean filter pre-processing method was used in the detection method presented in [57]. The operating principle of the technique is analogous to the median method. An example of a mean filter operation is shown in Figure 4.

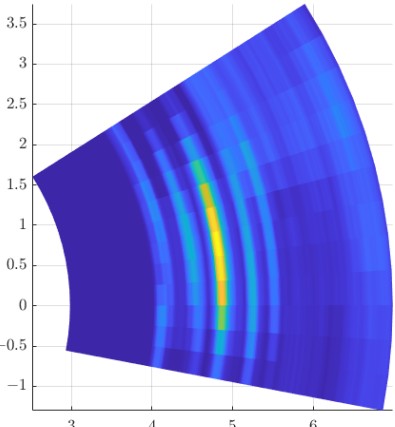

**Figure 4.** An example of mean filter application in pre-processing step. This method is based on ordering adjacent pixels and then applying the mean value for a specific filter size. The $5 \times 5$ pixels window was used for this processing case. [own source].

Authors of [58] present a comparison of the effects of the median, mean, wavelet-based, and morphological smoothing methods. The wavelet-based operation uses wavelet transformation for filtering the noise in the signal by splitting into the different scale (e.g., frequency) components [59]. The morphological smoothing method is based on erosion or dilation operations, which are more often used during the morphological processing stage. It reduces noise in the image obtained during the detection process. In a result of comparing the processing time, the obtained effect, the peak signal to noise ratio (PSNR), and the mean square error (MSE) of the above methods, the authors concluded that the most optimal methods are the median and mean methods.

As part of the pre-processing step, the scanning sector or region of interest can also be specified by selecting the distance range and the angular range of the area to be later segmented [60] (Figure 5). This reduces the amount of data processed in further image processing steps. This operation shortens the image processing time and is conducive to achieving real-time ODS operation.

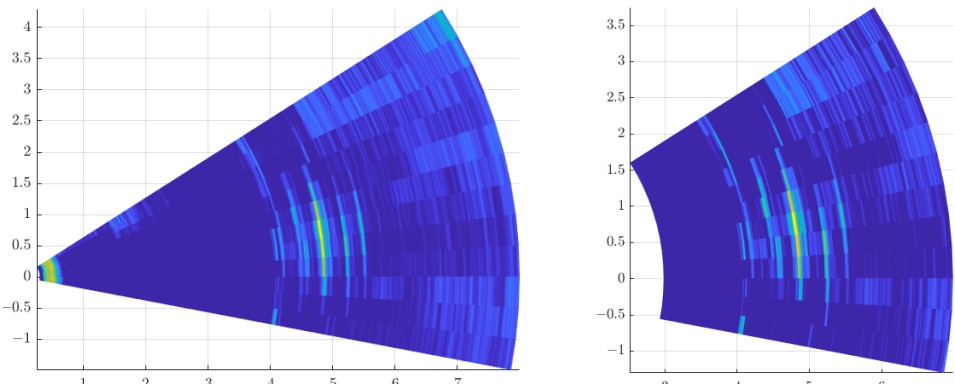

**Figure 5.** An example of determining ROI. This pre-processing technique reduces image area for faster image processing. [own source].

*3.2. Image Segmentation*

Image segmentation consists of creating classes of objects and qualifying individual pixels of the processed image. For example, in mine detection systems, classes of objects can be a reflection from the object, shadow, and background [61]. The main groups of segmentation methods are threshold segmentation, clustering, and Markov random fields method.

Threshold segmentation methods are based on comparing individual pixel values with a set threshold value. Based on that comparison, the pixel value is set to a specific value (0 or 1). In literature, modifications and improvements to this method can be found, such as Otsu threshold [62]. The thresholding operation with the gradient operator was presented in [63] to the vision-based image segmentation by searching the edge between areas.

Cluster analysis consists of classifying points into subgroups containing an appropriate degree of similarity. The purpose of segmentation based on cluster analysis is to distinguish such objects as echo, shadow, reflections, etc. Among the methods based on clustering, it can be distinguished by the K-means algorithm or region growing method. The K-means clustering technique [64] is based on determining K random points in the image and then assigning the closest points to each of them. Then the centroid of each cluster is calculated. Over time, the algorithm has been improved and modified. For example, [65] introduced the K-means clustering algorithm in conjunction with mathematical morphology. Another method is the region growing technique which is iterative checking of neighboring pixels and comparing their values with the averaged local value. The point is assigned to the region when the difference does not exceed the specified value. This method was used in [33] in the online processing framework for FLS images.

Markov random field is a method based on probability analysis of connections between adjacent pixels [66]. E.g., for an image obtained from SSS, if the pixel is close to the shadow, the probability that it belongs to it increases [61]. Various Markovian models for segmentation of sonar-based detection were presented in [37,67].

Figure 6 shows an example of the operation of the threshold segmentation algorithm. In Figure 7, threshold segmentation was preceded by mean filtering. It is worth noting that the use of the pre-processing method has a significant impact on the subsequent operation of segmentation algorithms.

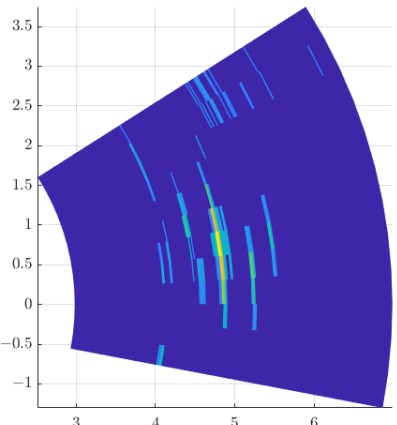

**Figure 6.** An example of threshold segmentation operation in sonar image processing. In this case, the application of threshold segmentation was not preceded by any pre-processing operation (except for determining the ROI). Therefore, many pixel groups can be seen as a result of this operation [own source].

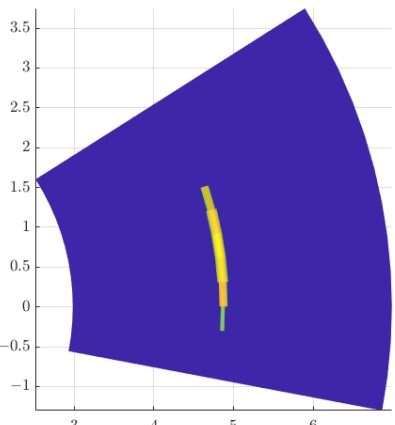

**Figure 7.** An example of mean filtering and threshold segmentation operations in sonar image processing. Due to the application of the mean filtering method before threshold segmentation, removing false positives related to reflections and noise have been achieved. The example above shows the importance of pre-processing operations for better image processing results.[own source].

### 3.3. Image Morphological Operations

Image morphology operations aim to improve the features of detected objects resulting from segmentation imperfections [68]. Basic operations in this step are dilation, erosion, opening, closing, and edge detection. Dilation operation is the expansion of an image of an object shape. It removes irregularities in the object's shape by extending its surface by the number of pixels depending on the structuring element (e.g., $3 \times 3$ or $5 \times 5$ pixels window). Erosion operation reduces the area of the object in the image as a result of comparison with the structuring element. In addition, image processing also uses other operations, which may be a combination of both of the above (opening, closing) or differing in how the structuring element affects individual pixels (skeletonization). After adjusting the image, it can be detected for such features as, e.g., boundary, edge, or point, depending on the application (Figures 8 and 9). They usually work by looking for significant value differences between neighboring pixels and marking them as an edge.

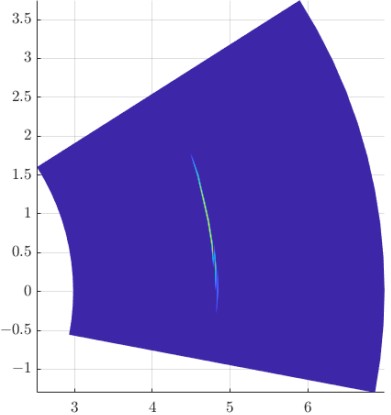

**Figure 8.** An example of edge detection operation in sonar image processing. The measurement was performed in a pool in which a cube-shaped object was placed. The side wall of the cube was directed perpendicular to the axis of the beam generated by the sonar. [own source].

### 3.4. Summary

Traditional image processing methods ensure adequate reduction of noise and interference generated in the sensor's perception of the environment. The result of such processing is detailed information about objects/obstacles near the AUV. Due to the fact that most techniques require checking the value of each pixel and subjecting them to mathematical or

statistical operations, they often require a large amount of computation: the more complex the method, the greater the processing time. Additionally, the author's experience in analysing and interpreting images is necessary for the methods to be properly tuned [55].

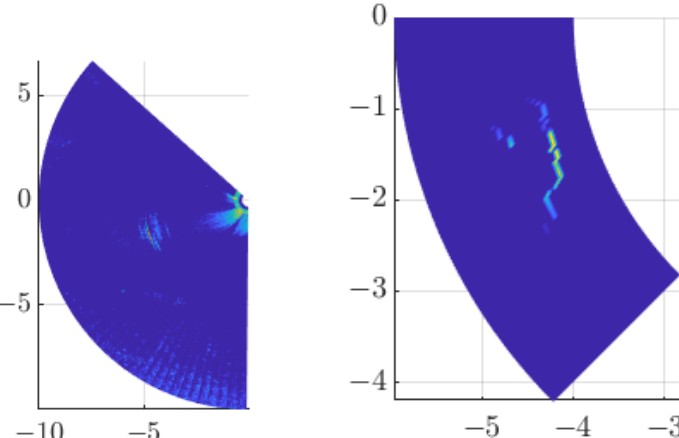

**Figure 9.** An example of image processing for sonar imagery measured at Lake Wysockie near Gdańsk (Poland) for the cylinder-shaped object. In measurements performed in a real environment, the impact of environmental instability on the final result of image processing can be noticed [own source].

## 4. ODS in Practically Tested AUV Capable of Collision Avoidance

This chapter discusses the obstacle detection approaches implemented in AUVs with obstacle avoidance and path planning capabilities. Based on the literature review presented in [2], only studies demonstrating path planning and collision avoidance algorithms tested practically in a real-world environment were selected for further analysis. Each selected research was analysed in detail regarding the equipment used to perceive the environment and the image processing operations. In addition, the solutions were assessed in terms of: complexity of the environment, static and dynamic obstacles detection ability, operational speed, and path planning suitability.

The operation of the vehicle named NPS ARIES [69] starts with the identification of the bottom. Then the ROI is determined. Once obstacle detected, the information about the obstacle's distance, height, and the centroid is sent to the controller. In image processing, a binary image is first created using a threshold. Then, during the erosion process, the value of each pixel in the $3 \times 3$ window is set to the minimum value. In the next step, the algorithm searches for the linear features of the object. Based on that operation, the position of the bottom is determined. The obstacle is identified using a Kalman Filter based on the vehicle pitch, pitch rate, and rotation angle. In the process of segmentation, areas forming lines are separated. The most significant areas are treated as potential obstacles. On this basis, the ROI is determined. Then, using the binary image, the contours of the obstacles are detected, which are tracked using the Kalman Filter. The method is effective and, together with appropriate path-planning algorithms, can provide optimal collision avoidance maneuvers. However, it is difficult to assess its effectiveness in a complex environment with more obstacles because the algorithm has been tested in a low-complication environment. In [70,71] no imaging methods were applied. The ODS works with the obstacle distance data obtained from the echosounders. Thanks to this, it was operated effectively in missions conducted by the authors. However, it does not guarantee effective operation in complicated conditions. Additionally, the AUV's obstacle avoidance maneuvers may not be optimal.

In reference [72], threshold and median filtering were first applied in the image processing, and then in the morphology step, erosion, dilation, and edge detection methods were used. In the experiment, the vehicle correctly avoided the breakwater obstacle. It proves the correct operation of the obstacle detection method, but the environment in which

the experiment was performed was not complex. In reference [73], the vehicle's operation in the underwater environment with the presence of obstacles was tested. The obstacle detection method is based on measuring the distance to an object using a collision sensor or proximity sensor and is effective for static obstacles. The authors [74] presented a solution based on three echosounders for measuring the distance to obstacles in front of the vehicle. In the study [75], five echosounders were used in the ODS for octree space representation. Obstacle detection systems that use only echosounders to measure the distance to an obstacle are prone to interference. In addition, it does not allow for optimal collision avoidance maneuvers. In research [76], sonar imaging was used in the obstacle detection system, which is then filtered using mean and median filtering. In the segmentation step, a fuzzy K-mean clustering algorithm is used. In morphological processing, the occupancy map is determined from the grayscale image. The method ensures sufficient accuracy and, after applying appropriate algorithms, allows for near-optimal path planning. In reference [77], the sonar image is first speckle noise suppression by a $17 \times 17$ filter, then the local image histogram entropy method ($9 \times 9$) is used. In the next step the hysteretic threshold of entropy is applied to the image. Finally, the edge detection process is performed, and the obstacles are saved on the map. The method allows making decisions in near real-time and planning near the optimal path. The study [18] presented a vehicle capable of avoiding a collision. An experiment confirmed the correct operation in a real underwater environment. However, the obstacles were simulated, which allowed for bypassing the detection process and the related problems. Therefore, this study will not be considered for further analysis. The authors of [78] presented a solution based on two sets of line laser and camera. The image obtained from the sensors in this configuration is subjected to top-hat transformation based on opening and subtraction operations through the $1 \times 21$ pixels window to remove bright points from the background of the image (the background is not completely black). Then the image is binarized by the threshold value. White boxes are annotated using the fast label method, and groups of less than 80 pixels are removed as noise. An obstacle is identified if it is present in 5 or more frames. The method is effective, but it has a small operating range, and the processing time does not allow the vehicle to be controlled in a complex environment close to real-time. The same solution with an additional sensor in the form of FLS was presented in [49]. By that implementation, the authors obtained a greater range of ODS activities. In reference [79], two sonars were used to provide 210 degrees FOV. The vehicle has the ability to follow the wall parallel to the AUV axis. The vehicle will perform avoidance maneuvers depending on the sector in which the obstacle appears. The system detects an obstacle when it is present on five or more returns or when the tracked wall is in front of the AUV. The method does not focus on the features of the obstacle and its size. Therefore it does not allow for the determination of the optimal route and the effective movement of the AUV in a complex underwater environment. In [80], the echo intensity matrix obtained from sonar is filtered by the specified threshold method. Then a range is computed for points with intensity greater than the threshold, representing the distance to the obstacle. Extensive AUV experiments were carried out in various scenarios, confirming ODS's effectiveness. By the use of appropriate path planning algorithms, the AUV has the ability to move near the optimal path. In reference [81] detection is based on the point cloud, the parameters of which in space are estimated using scaling factors based on depth and inertial measurement unit (IMU) measurements. In a pre-processing step, the contrast adjustment is performed along with histogram equalization. The SVIn method [39] outputs a representation of the sensed environment as a 3D point cloud, which is later subjected to extracting visual objects with a high density of features. The method uses the density-based clustering operation in the segmentation step. After clusters detection, their centroids are determined. The method allows us to determine the near-optimal path and operate in real-time. In the study [82], a threshold-based operation for segmentation was first performed in sonar image processing. Then edge detection is applied. The method provides real-time obstacle detection and can be used to move in the underwater environment in the presence of

both static and dynamic obstacles. Reference [83] presents ODS based on DL methods for fishing net detection. Pre-processing of the FLS images is conducted by gray stretching and threshold operations. Authors trained and tested their network to achieve ATR. Learning of MRF-Net based on the data collected in the sea and mix-up strategy based on using randomly synthesized virtual data. As a result, the system detects and classifies an obstacle as a fishing net with very high accuracy. The method is very effective and allows the AUV to operate in real-time. However, applying the technique to other obstacles must be preceded by a learning process based on images of the specific obstacle. In reference [84], vision images are used in ODS. First, image features such as intensity, color, contrast, and light transmission contrast are determined. The next step is the appropriate global contrast calculation. After that, ROI is detected, and threshold-based segmentation is executed. AUV successfully detected and avoided obstacles in a complex environment in pool tests. The method allows for near real-time operation. The disadvantage of this method is the range which depends on the visibility. All the above methods are chronologically presented in Table 2 with information about the hardware used, main image processing properties, and effectiveness evaluation.

**Table 2.** Evaluation of AUVs Obstacle Detection Systems capable of collision avoidance.

| Source | Year | Hardware Used in ODS | Effectiveness | Main Properties |
|---|---|---|---|---|
| [69] | 2005 | Forward looking sonar | Low | • Sonar resolution $491 \times 198$<br>• Binary image by threshold<br>• Erosion<br>• Determination of the bottom slope<br>• Feature extraction<br>• Determination of ROI<br>• Contours of obstacles and surface calculation<br>• Kalman filter tracking |
| [70,71] | 2008 | Echosounder | Low | • Margin for detection depends on the AUV speed<br>• Filtering out a false reflection from the surface<br>• Indication of possible surface icing during ascent<br>• Limited headroom indication based on comparison of the echosounder data with data from other sensors (immersion, inclination) |
| [72] | 2009 | Blazed array multibeam sonar | Medium | • Pre-processing by specifying a background threshold level<br>• Median filtering<br>• Morphology: erosion, dilation, edge detection |
| [73] | 2011 | Vision camera, imaging sonar, echosounder, side scan sonar | Low | • Detection based on collision sonar or proximity sensor<br>• Detection of obstacles at a distance shorter than the specified activate obstacle avoidance algorithms |
| [74] | 2014 | 3 single beam ranging sonar | Medium | • Bottom tracking with an echosounder<br>• Avoidance of noise from cooperating devices using the ping synchronization scheme<br>• Threshold segmentation |
| [75] | 2015 | 5 echosounders | Medium | • Octree-based representation of points in space based on data obtained from echosounders working as a single beam sensor |
| [76] | 2015 | Multibeam sonar | High | • Filtering: median filtering, mean filtering<br>• Segmentation: fuzzy K-mean clustering algorithm<br>• Morphological processing—binarization |

**Table 2.** *Cont.*

| Source | Year | Hardware Used in ODS | Effectiveness | Main Properties |
|---|---|---|---|---|
| [77] | 2016 | 2 Forward-looking sonars | High | • Speckle noise suppression<br>• Local image histogram entropy<br>• Hysteretic threshold of entropy<br>• Feature extraction—Edge detection |
| [18] | 2016 | Obstacles are simulated | - | • No obstacle detection system<br>• Real-world tests of only collision avoidance and path planning algorithms without obstacle detection consideration |
| [78] | 2016 | 2 cameras<br>2 line lasers | Low | • Morphological filter to extract features, uneven luminance<br>• Top hat transformation (opening, subtraction)<br>• Binarization by Threshold<br>• Description by fast label method<br>• Removing groups of pixels below 80<br>• A decision about the classification of an obstacle based on five or more images |
| [49] | 2018 | Forward looking sonar<br>2 cameras<br>2 line lasers | Medium | • Solution based on the sensors and algorithms used in [78]<br>• Added FLS to increase the range of ODS operation |
| [79] | 2018 | 2 forward-looking sonars | Medium | • 210 deg FOV<br>• The system detects an obstacle when it takes five or more returns or when the tracked wall is in front of the AUV |
| [80] | 2019 | Mechanical scanning imaging sonar,<br>4 echosounders<br>3 cameras | High | • Threshold segmentation<br>• Real-time operating<br>• Extensive numerical and practical tests executed |
| [81] | 2020 | 3 cameras | High | • SVIn2 vision inertial state estimation system based on visual data augmented with IMU sensor data, which are linked together in a visual form<br>• Histogram equalization for contrast adjustment<br>• Extracting visual objects with a high density of features from a point cloud |
| [83] | 2021 | Multibeam forward-looking sonar | High | • Gray stretching and threshold segmentation<br>• Normalization methods<br>• Deep reinforcement learning for proper obstacle detection and avoidance<br>• Mixup learning strategy |
| [84] | 2021 | Camera | High | • Determination of image features such as intensity, color, contrast, and light transmission contrast<br>• Appropriate global contrast calculation<br>• ROI detection<br>• Threshold-based segmentation |
| [82] | 2021 | Multibeam echosounder forward-looking sonar | Medium | • Threshold segmentation method<br>• Edge detection of the object |

## 5. Analysis, Bottlenecks, Future Works

Section 4 describes the obstacle detection and image processing systems for AUVs capable of avoiding obstacles, the correct operation of which has been confirmed by tests

performed in a real-world environment. This chapter intends to summarize the results of the analysis carried out in the previous chapter. Statistics on the evaluation of the effectiveness of ODS depending on the research's publication time are presented together with analysis of the equipment used to perceive the environment in ODS. Additionally, the number of research corresponding with low, medium, and high effectiveness are shown. Then the limitations that inhibit the development of ODSs and future work in this field are defined.

### 5.1. Analysis

After examining the literature of ODS in AUV, it can be seen that there is a constant development in this field. In Figure 10, the effectiveness of each considered system is evaluated. The surveys are arranged in chronological order. The efficiency has been assessed as low, medium or high depending on below parameters:

1. Complexity of the environment in which the AUV was tested. Each article was rated on a scale of 0–3 depending on the number of obstacles included in the tested environment, the distance between them, the ability to detect obstacles with irregular or only simple shapes, and whether the vehicle was tested for 2D or 3D maneuvers.
2. Ability to only static or static and dynamic obstacles detection. The studies were rated on a 0–1 scale, depending on whether the presented solution can operate correctly in the presence of static and dynamic or only static obstacles.
3. ODS operation speed. This parameter was assessed based on data such as sonar update rate, frame rate, path replanning time, and other specific parameters of the analysed systems. The solutions were rated on a 0–2 scale, where 1 Hz was chosen as the reference frequency of the environment detection and image processing procedure. However, in some studies, the ODS operation speed assessment was based on estimated values because of limited information about the system.
4. Suitability for path planning which is the potential ability to provide sufficient data to determine the optimal path. The research was rated on a 0–3 scale depending on the image processing methods used, the accuracy of presenting obstacles after image processing, and the optimality of the executed path during tests in a real-world environment.

The sum of points determined the final evaluation where high efficiency was in the range of 7–9, medium 4–6, and low 0–3. Detailed assessment data are provided in Table S1.

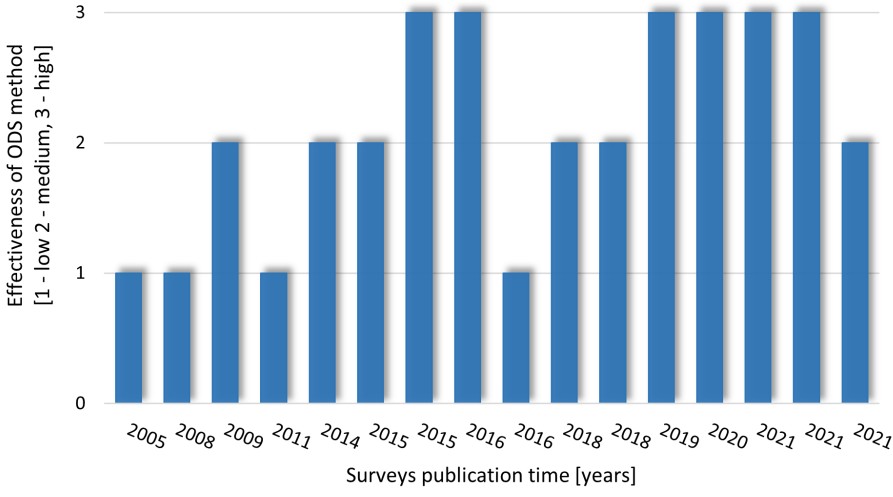

**Figure 10.** Effectiveness of ODS in time, where 1 is low, 2 is medium and 3 is high efficiency. The constant development of ODS is noticeable, caused mainly by increasing the computing power of control systems.

It can be seen that at the initial studies, the effectiveness of the methods was low. In the first decade of the 21st century, the main challenge was to obtain a high speed of operation, allowing the AUV to avoid collisions in real-time. The situation was related to the limitations of the large number of calculations that must be performed as part of image processing. In addition, the first designs of AUVs with collision avoidance algorithms did not focus on determining the optimal route. Hence, the first ODSs were not advanced in determining the characteristics of the obstacles found in the environment where the AUV was tested. The ODSs were refined over time. For example, in references [49,78], Forward-looking sonar was added to the initially developed system based on forward-looking cameras and line lasers to increase the obstacle detection range. In recent times, along with technological development and the increase in computational resources, researchers have used more advanced systems that ensure fast data processing, operate in a complex environment, and enable the determination of a near-optimal path. Additionally, in reference [83], the method of AI based on deep learning was used to detect obstacles. The authors presented the effectiveness of this method in detecting the fishing net.

In terms of environment perception sensors, researchers most often used sonar imagery (Figure 11). Hydroacoustic wave-based sensors provide the greatest range. This directly increases the time the AUV has to react to an obstacle on its path. However, sonar imagery requires appropriate analysis experience and proper operations to suppress background noise. Additional limitations in using sonars are false positives resulting from the reflection of the signal from the water surface and the bottom. For the above reasons, some researchers decided to base on the visual image from the camera. It provides more details related to contrast, texture, and color. However, the vision system range is small due to the high attenuation of light in the underwater environment. Some studies used a combination of several sensors [49,73,80], which increases the amount of data to process.

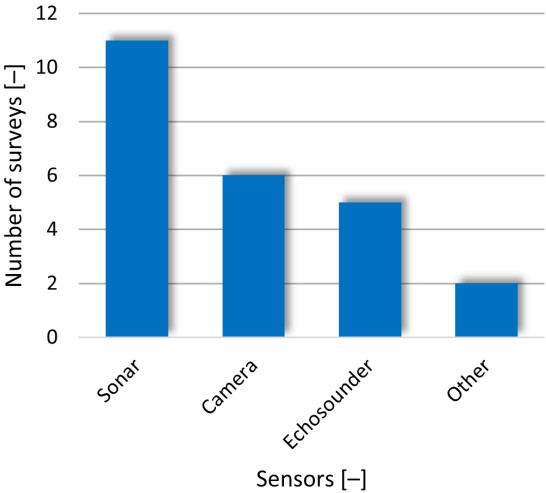

**Figure 11.** The number of surveys depending on the sensor used to perceive the environment. Due to the low attenuation level of acoustic signals in an underwater environment, sonar-based solutions are the most popular device chosen in ODS.

In Figure 12 the analysed literature is presented in three groups of effectiveness. The number of ODSs with low, medium, and high efficiency is almost the same. This proves the constant development of systems. It is most likely that further developed methods will be highly effective. It also indicates that the ODS field is less challenging than 10–15 years ago.

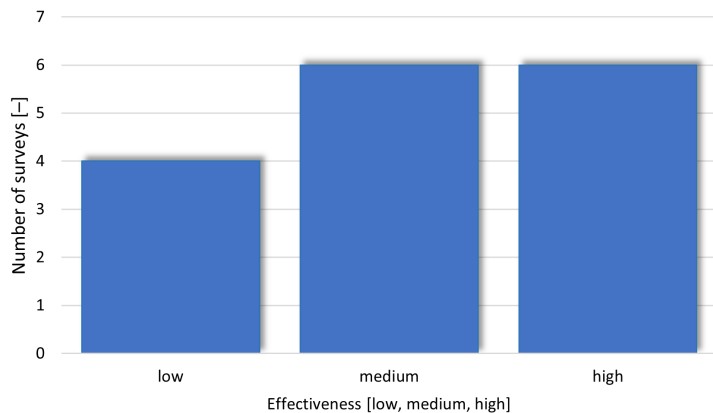

**Figure 12.** The number of surveys depending on the ODS effectiveness. Almost the same number of low, medium, and high-efficiency solutions proves the constant development of ODSs.

### *5.2. Bottlenecks and Future Works*

Currently, researchers dealing with obstacle detection for AUVs field have mastered many issues related to the demanding underwater environment. Developed ODSs work in close to real-time and ensure adequate accuracy for determining the near-optimal path. However, the results accuracy strictly depends on the AUV speed. In the studies analyzed in this paper, more complex methods allow AUVs to develop a speed of several meters per second. Increasing the speed is associated with increasing the range of ODS operation. For the camera-based solutions, the range is limited by light attenuation and is usually several meters. In the case of sonar technology, the greater range is associated with, the longer time the signal needs to irradiate the space. More data is related to the number of samples depending on the distance resolution. Reducing the resolution at a distance may reduce the amount of data , however, it will also reduce the accuracy of the ODS. If the obstacle is small enough, it may not be registered, or the receiver may treat obstacles close to each other as one. It is also worth noting that in the above analysis, almost all methods rely on the classic image processing approach. Artificial intelligence methods have been extensively developed in recent years. Nevertheless, they are rarely used in solutions for AUVs capable of collision avoidance. This may be due to certain limitations in expert knowledge or the lack of an adequate amount of data needed for training (for machine learning methods). Sonar or video images of the underwater environment are not as widespread as for other environments. This practically requires extensive measurements by researchers, which is usually time-consuming. The effectiveness of the methods is highly dependent on the amount of training materials. For this reason, virtually synthesized images can be a promising solution.

### 6. Conclusions

This article presents an in-depth analysis of the ODSs for collision avoidance AUVs, including a basic explanation of underwater perception sensors, image processing and a detailed description of the ODSs tested under real conditions. The underwater environment is very challenging in perception. For this reason, developing an effective system that can enable optimal and safe path planning requires implementing specific solutions. Nevertheless, the above analysis results show that researchers have well mastered this part of underwater robotics. The recently developed systems are more and more effective. In the future, even better ODSs parameters could be achieved by using AI methods.

**Supplementary Materials:** The following supporting information can be downloaded at:https://www.mdpi.com/article/10.3390/electronics11213615/S1, Table S1: Assessment of analysed papers.

**Funding:** This research received no external funding.

**Institutional Review Board Statement:** Not applicable.

**Informed Consent Statement:** Not applicable.

**Data Availability Statement:** Not applicable.

**Conflicts of Interest:** The authors declare no conflict of interest.

## Abbreviations

The following abbreviations are used in this manuscript:

| | |
|---|---|
| AI | Artificial Intelligence |
| ATR | Automatic Target Recognition |
| AUVs | Autonomous Underwater Vehicles |
| AVs | Autonomous Vehicles |
| CNN | Convolutional Neural Network |
| DL | Deep Learning |
| DOA | Direction Of Arrival |
| FLS | Forward Looking Sonar |
| FOV | Field Of View |
| IMU | Inertial Measurement Unit |
| MBES | Multibeam Echosounder |
| MSS | Mechanical Scanning Sonar |
| ODSs | Obstacle Detection Systems |
| ROI | Region Of Interest |
| ROVs | Remotely Operated Vehicles |
| SBES | Single beam Echosounder |
| SLAM | Simultaneous Localization and Mapping |
| SNR | Signal to Noise Ratio |
| SSS | Side Scan Sonar |
| UASs | Unmanned Aerial Systems |
| UAVs | Unmanned Aerial Vehicles |
| UGVs | Unmanned Ground Vehicles |
| UUVs | Unnmaned Underwater Vehicles |

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
