# Peer review of "Review of Obstacle Detection Systems for Collision Avoidance of Autonomous Underwater Vehicles Tested in a Real Environment"

_electronics, doi:10.3390/electronics11213615_

Round 1

Reviewer 1 Report

This paper presents review of underwater obstacle detection systems (ODS), which indicate the realized systems integrating hardware and software. The author’s previous review paper (Ref. [2]) dealt with underwater navigation algorithms. I wonder if the review of realized ODS is necessary. Algorithms for collision avoidance and path planning have been reviewed in many articles, including the author’s article. At first glance, this paper seems to be focused on hardware. Section 2 deals with various sensors used in ODS. However, Section 3 introduces image processing algorithms, and Section 4 evaluates a lot of ODS. It is unclear what this article focuses on. If this article is focused on realized ODS hardware, it would be better to include pictures of various ODS instead of sonar images. Too many sonar images are included in this article. In Table 1 and Figure 8, effectiveness of ODS is evaluated using the three levels, high, medium, and low. However, criteria or method for the evaluation is not described. Thus, the evaluation is not reliable.

This article has many grammatical and typographical errors, for instance:

- Only some of the words in the title care capitalized with the first letter.

- The first letter of the keywords should be capitalized.

- Obstacle Detection System (ODS) -> obstacle detection system (ODS).

- Undefined acronyms are used (e.g., UUV).

- pith -> pitch, pith rate -> pitch rate (Page 10)

All figures should be located at top, but Figs. 2 and 3 do not.

There are empty blanks at the bottom of pages 8 and 9.

Too many conference papers are included in the references.

In summary, this article is not carefully written, and its contribution is unclear.

Reviewer 2 Report

The main content of this work is a review of obstacle detection systems for collision avoidance of autonomous underwater vehicles, and the sensors used in the collision avoidance obstacle detection system are introduced. In this paper, various existing obstacle detection systems are evaluated in detail, including the effectiveness of obstacle detection algorithms, and the characteristics of each obstacle detection system are compared and summarized in the form of a table. Finally, the bottleneck of this kind of work and the future research direction are pointed out. The detailed audit opinions are as follows:

(1) It is suggested to check the full text and correct the grammatical errors in the article.

(2) Pictures are lack of detailed explanation, please add.

(3) The introduction is more like a background description and does not highlight the significance of this work.

(4) There are few data in the past five years, so it is suggested to supplement the high-quality literature in the past five years.

(5) It is suggested to supplement the specific workflow of ODS from detecting obstacles to avoiding obstacles. It is suggested to supplement the comparison of more measured data.

(6) It is suggested that the author compare the existing ODS performance parameters.

Round 2

Reviewer 1 Report

The paper has improved, but there still remain things to be improved. The contribution of this paper should be described in detail in the introduction part. In addition, the detailed process and method for evaluation of ODS should be described in Section 5. 

Reviewer 2 Report

Thank you for carefully addressing the comments raised in the first round of review. The quality of this manuscript has significantly improved. I support the publication of this manuscript with no further comments on its contents.

Author Response

Dear Reviewer,

Thank you for acceptance of this manuscript.  I am very grateful for your time and valuable comments.

Sincerely,

Rafał Kot

Polish Naval Academy

Round 3

Reviewer 1 Report

I have no further comments.